# IL-6 trans-Signaling Impairs Sprouting Angiogenesis by Inhibiting Migration, Proliferation and Tube Formation of Human Endothelial Cells

**DOI:** 10.3390/cells9061414

**Published:** 2020-06-05

**Authors:** Mulugeta M Zegeye, Blanka Andersson, Allan Sirsjö, Liza U Ljungberg

**Affiliations:** 1Cardiovascular Research Center, School of Medical Sciences, Örebro University Södra Grev Rosengatan 32, 703 62 Örebro, Sweden; allan.sirsjo@oru.se (A.S.); liza.ljungberg@oru.se (L.U.L.); 2Department of Biomedical and Clinical Sciences (BKV), Division of Inflammation and Infection (II), Linköping University Campus US, 581 85 Linköping, Sweden; blanka.andersson@liu.se

**Keywords:** VEGF-A signaling, neoangiogenesis, cytokine, Matrigel, HUVECs, IPA

## Abstract

Sprouting angiogenesis is the formation of new capillaries from existing vessels in response to tissue hypoxia due to growth/development, repair/healing, and also chronic inflammation. In this study, we aimed to elucidate the effect of IL-6, a pleiotropic cytokine with both pro-inflammatory and anti-inflammatory functions, in regulating the sprouting angiogenic response of endothelial cells (ECs). We found that activation of IL-6 trans-signaling inhibited the migration, proliferation, and tube formation ability of ECs. In addition, inhibition of the autocrine IL-6 classic-signaling by depleting endogenous IL-6 from ECs impaired their tube formation ability. At the molecular level, we found that IL-6 trans-signaling in ECs upregulated established endogenous anti-angiogenic factors such as *CXCL10* and *SERPINF1* while at the same time downregulated known endogenous pro-angiogenic factors such as *cKIT* and *CXCL8*. Furthermore, prior activation of ECs by IL-6 trans-signaling alters their response to vascular endothelial growth factor-A (VEGF-A), causing an increased p38, but decreased Erk1/2 phosphorylation. Collectively, our data demonstrated the dual facets of IL-6 in regulating the sprouting angiogenic function of ECs. In addition, we shed light on molecular mechanisms behind the IL-6 trans-signaling mediated impairment of endothelial sprouting angiogenic response.

## 1. Introduction

Sprouting angiogenesis is the growth of new capillaries from pre-existing vessels in response to tissue hypoxia due to growth, development, and repair [1]. It involves degradation of the extracellular matrix, migration, differentiation, and proliferation of endothelial cells (ECs) [1,2]. This multi-step process is tightly regulated by a complex interaction of pro- and anti-angiogenic factors which are normally balanced in the adult organism. However, tissue hypoxia induces secretion of the vascular endothelial growth factor-A (VEGF-A) which is the dominant and rate-limiting activator of angiogenesis [3,4]. VEGF-A binds to its receptor vascular endothelial growth factor receptor-2 (VEGFR-2) expressed on ECs that, upon ligand binding, dimerizes and trans-phosphorylates tyrosine residues in its intracellular domain [5]. It subsequently provokes downstream signaling pathways including PI3K/Akt, p38/MAPK, and pErk/MAPK pathways that regulate survival/permeability, migration, and proliferation [6,7,8,9].

Aberrant angiogenesis has been implicated to exacerbate various chronic inflammatory conditions including atherosclerosis, rheumatoid arthritis, psoriasis, and cancer, as the newly formed vessels usually are leaky and unstable [10,11,12,13]. Pro-inflammatory mediators such as TNF-α and IL-1 trigger angiogenesis through induction of endothelial tip cell formation and proliferation, or via increased recruitment of inflammatory cells such as macrophages that serve as a source of VEGF-A [14,15].

The pleiotropic cytokine IL-6, which is secreted by various cell types including human vascular ECs, has both pro-inflammatory [16] and anti-inflammatory [17,18,19] functions. These contrasting functions have been proposed as being dependent on how IL-6 interacts with target cells and also the signaling pathways it provokes downstream. IL-6 signaling requires a heteromeric receptor composed of IL-6R and gp130 signal transducer. The gp130 signal transducer is ubiquitously expressed in all types of cells while the membrane-bound IL-6R is found only on a few types of cells, including hepatocytes and immune cells [20]. However, there also exists a soluble form of IL-6R (sIL-6R) in body fluids, predominantly as a result of proteolytic shedding [21]. Hence, IL-6 can either bind to a membrane-bound IL-6R or to a soluble form of IL-6R (sIL-6R) that subsequently engage the signal transducing receptor gp130 to elicit classic- or trans-signaling pathways, respectively. Activation of the IL-6 classic signaling pathways has been proposed to have regenerative/homeostatic and anti-inflammatory functions while trans-signaling leads to pro-inflammatory responses [22]. Although previous studies have shown an indirect pro-angiogenic activity promoted by IL-6 [23,24,25,26], its direct impact in regulating endothelial angiogenic response has not been investigated. In addition, we previously reported that vascular ECs express the membrane-bound IL-6R and gp130 that render these cells responsive to IL-6 classic-signaling, the angiogenic significance of which has largely been overlooked [27]. Hence, in this study, we aimed to unravel the role of IL-6 classic- and trans-signaling pathways in regulating the angiogenic function of ECs.

## 2. Materials and Methods

### 2.1. Cell Culturing

Human umbilical vein endothelial cells (HUVECs, Life Technologies, USA) were cultured using complete endothelial medium (VascuLife basal medium supplemented with VEGF LifeFactors kit (LifeLine Cell Technologies, Carlsbad, USA)) and antibiotics (penicillin (0.1 U/mL) + streptomycin (100 ng/mL)-PEST, Gibco, Life Technologies, USA). The T-75 flasks (Sarstedt, Nümbrecht, Germany) containing the cells were incubated at 37 °C in a 5% CO_2_ environment. By replacing medium every 48–72 h and/or sub-culturing upon confluence, the cells were kept until passage 10.

### 2.2. Stimulation of HUVECs with IL-6 and sIL-6R

HUVECs (3 × 10^5^ cells per well for 6-well plates and 6 × 10^4^ cells per well for 24-well plates) were cultured overnight in complete endothelial medium containing antibiotics. The following day, the medium was replaced with fresh antibiotic-free medium and the cells were treated with IL-6 (100 ng/mL) and sIL-6R (100 ng/mL) (both from R&D systems, Minneapolis, USA). For stimulation of IL-6/sIL-6R pretreated cells with VEGF-A (10 ng/mL; VEGF165, R&D systems, Minneapolis, USA), the cells were rinsed, and the medium was replaced with growth factor reduced Opti-MEM (Gibco, Life Technologies, Carlsbad, USA). At the end of incubations, supernatants were collected and kept at −80°C until further analysis. The cells were either stored at −80°C until further analysis or detached with trypsin-EDTA to be used for tube formation assay.

### 2.3. IL-6 Knockdown Using siRNAs

In 6-well plates, 2 × 10^5^ HUVECs per well were seeded in complete endothelial medium with antibiotics and incubated overnight. The cells were rinsed with growth factor reduced opti-MEM (Gibco, Life Technologies, Carlsbad, USA), and 700μL Opti-MEM containing 4μL lipofectamine 2000 (Invitrogen, Carlsbad, USA), and a mix of 3 stealth siRNAs targeting IL-6 (10nM of each siRNA, Invitrogen, Carlsbad, USA) was added into each well. The control wells instead had non-target siRNAs (30nM, medium GC content, Invitrogen, Carlsbad, USA). After 4 h of incubation, 1.3 mL of complete endothelial medium was added into each well, and incubation continued for 48h. At the end, the culture supernatants were collected and stored at −80 °C until further analyses. The cells were rinsed with PBS and detached with trypsin-EDTA to be used for tube formation assay.

### 2.4. Tube Formation Assay

Frozen vials of Matrigel Membrane Matrix (Fisher Scientific, USA) were thawed on ice at +4 °C overnight. Matrigel was added (50 µL/well) into 96-well plates (TPP Techno Plastic Products AG, Trasadingen, Switzerland) and allowed to polymerize at 37 °C for 30 min. Meanwhile, HUVECs were detached from wells/flasks using trypsin-EDTA (Gibco, Life Technologies, Carlsbad, USA). Using basal Vasculife medium (2% Fetal Bovine Serum/FBS) with or without IL-6/sIL-6R (100 ng/mL each), suspension of HUVECs was prepared, and 10^4^ cells per well were added to the polymerized Matrigel in duplicate. The plates were placed in IncuCyte S3 Live Cell Analyses System (Sartorius AG, Göttingen, Germany) and the tube formation progress was monitored over a period of 24 h by taking pictures every hour (3 images per well). The images (10× magnification) were subsequently analyzed using Angiogenesis-Analyzer macro written for ImageJ1. The macro is available online at http://imagej.nih.gov/ij/macros/toolsets/Angiogenesis%20Analyzer.txt.

### 2.5. Proliferation Assay Using Crystal Violet Staining

HUVECs were seeded in 24-well plates (2 × 10^5^ cells per well) containing 0.5 mL complete endothelial medium with antibiotics, and were incubated overnight. The next day, 1 plate was fixed and stained to be used as the number of cells at baseline (time = 0 h). For the other plates (time = 24 h to 168 h), the medium was replaced every 72 h with antibiotic-free complete endothelial medium, and the cells were stimulated with IL-6 alone or in combination with sIL-6R. At the end of the incubation periods, the cells were fixed and stained with crystal violet solution. Briefly, the medium was removed and 500 µL of a solution containing 0.1% crystal violet in 20% methanol was added into each well, and the plate was then incubated on a shaker (200 cycles/min) for 20 min at room temperature. Then, the wells were washed 5 times using deionized water and were allowed to air-dry for 24 h. By adding 250 µL of 10% acetic acid solution and shaking the plate for 5 min, the dye retained by the cells was solubilized, and the optical density (OD) in the supernatant was measured at 590 nm using Cytation 3 Imaging reader (BioTek, Winooski, USA). The OD at time = 0 h was set to 1 and a ratio was calculated on the basis of the OD from time = 24 h to 168 h divided by the OD from time = 0 h.

### 2.6. Apoptosis Assay

Detection of apoptotic HUVECs was performed using FITC annexin V apoptosis detection kit with 7-aminoactinomycin D (7-AAD; BioLegend, United Kingdom) according to the manufacturer’s instructions. Briefly, HUVECs (3 × 10^5^ cells per well, 6-well plate) treated with IL-6 alone or in combination with sIL-6R for 48 h were detached with 1mM EDTA in PBS supplemented with 2% FBS. The cells were pelleted by centrifugation (230× *g* for 4 min at 4 °C) and resuspended in annexin V binding buffer. After the addition of FITC annexin V and 7-AAD viability staining solution, the cell suspensions were gently vortexed and incubated for 15 min at room temperature in the dark. More annexin V binding buffer was added into each cell suspension and the preparations were sorted using a Gallios Flow Cytometer (Beckman Coulter Life Sciences, Wycombe, United Kingdom) and analyzed using Kaluza flow cytometry analysis software version 1.3 (Beckman Coulter, Wycombe, United Kingdom). Cells that were negative for both annexin V and 7-AAD were considered as viable (non-apoptotic/non-necrotic) cells.

### 2.7. Migration Assay

HUVECs (6 × 10^4^ cells per well) were seeded in 24-well plates (Corning Inc., New York, USA) containing complete endothelial medium with PEST, and they were allowed to attach for 2 h. The medium was replaced with fresh anti-biotic free complete endothelial medium and the cells were stimulated with IL-6 and sIL-6R for 24 h. A scratch was made using a 200 µL pipette tip and the wells were rinsed followed by addition of fresh medium, and then IL-6/sIL-6R were added into the wells. Migration of the cells to cover the scratch was then monitored over a period of 12 h using IncuCyte S3 Live Cell Analyses System (Sartorius AG, Göttingen, Germany) by collecting images at 4× magnification every 4 h. The images were subsequently analyzed using MRI Wound Healing Tool macro written for ImageJ1, available online at http://dev.mri.cnrs.fr/projects/imagej-macros/wiki/Wound_Healing_Tool. Percentage of gap closure was calculated and compared between controls and treatment groups.

### 2.8. Total RNA Isolation and cDNA Synthesis

Following the manufacturer’s instructions, we extracted total RNA from frozen cells using E.Z.N.A Total RNA Kit I (OMEGA bio-tek inc, Norcross, USA). Briefly, cells were lysed with TRK lysis buffer, and the lysates were mixed with equal volume of 70% ethanol. The mixtures were transferred into HiBind RNA columns and centrifuged at 10,000× *g* for 1 min. The columns were washed 3 times with wash buffers and the RNA was eluted using RNase-free water. Using a NanoDrop 2000 (Thermo Fisher Scientific, Waltham, USA) spectrophotometer, we determined the RNA quantity and purity. The RNA extracts were used to synthesize cDNA using a high capacity cDNA reverse transcription kit (Thermo Fisher Scientific, Waltham, USA) according to the manufacturer’s instructions. Briefly, a mixture of 1μg RNA extract and master mix containing buffer, random primers, dNTPs, reverse transcriptase enzyme, and nuclease-free water adjusted to total volume of 20 µL was prepared for each extract. A negative control containing master mix and water instead of RNA was also included. The following setup was used for thermal cycling: 10 min at 25 °C, 120 min at 37 °C, and 5 min at 85 °C, and kept at 4 °C before storage at −20 °C.

### 2.9. Cell Lysate Preparation and Total Protein Quantification

After rinsing HUVECs with PBS, we added ice-cold RIPA lysis buffer (Millipore, Burlington, USA) to lyse the cells. A Micro BCA Protein Assay kit was used (Thermo Scientific, Waltham, USA) according to the manufacturer’s instructions to quantify the proteins in the cell lysates. Absorbance at 562 nm was measured using a Cytation 3 Imaging reader (BioTek, Winooski, USA).

### 2.10. Human Angiogenesis Array

Expression of angiogenesis-related genes was studied in 3 independent experiments using TaqMan Human Angiogenesis Array (Applied Biosystems, Foster City, USA). This array is pre-coated with 92 primers/probes targeting angiogenesis associated genes and 4 housekeeping genes. A mixture of TaqMan Fast Advanced Master Mix (2×, Applied Biosystems, Foster City, USA), cDNA, and nuclease-free water was added into each well (10 µL/well) and the plate was run in QuantStudio 7 Flex Realtime PCR system (Applied Biosystems, Foster City, USA). The cycling condition used was as follows: at 95 °C for 1 s and at 60 °C for 20 s for 40 cycles following one-step initialization at 50 °C and 95 °C for 2 min each. Then, Ct values were recorded for each gene and ΔΔCt values were calculated using the expression of GAPDH as a house keeping gene. Statistical analysis was performed using *t*-test followed by Benjamini–Hochberg false discovery rate (FDR) test for multiplicity correction. For further analyses, we employed Ingenuity Pathway Analysis (IPA, QIAGEN Inc., Venlo, The Netherlands, https://www.qiagenbioinformatics.com/products/ingenuity-pathway-analysis) online tool [28]. A core analysis was performed by setting a cut-off value for fold change (FC) and FDR to be 1.5% and 10%, respectively, in order to have a sufficient number of genes for the analysis. This gave us 37 analyses-ready genes (15 upregulated and 22 downregulated, Appendix A). Significantly enriched angiogenesis-related functions by these genes were identified using the “Grow” tool, and the activation status of these functions was determined by “Molecular Activity Predictor (MAP)” tool. The enriched functions “Neovascularization”, “Formation of endothelial tube”, “Tubulation of endothelial cells”, and “Vasculogenesis” are combined into one network referred to as “tube formation of ECs”. Similarly, the enriched functions “Cell movement of endothelial cells’ Migration of endothelial cells” and “Migration of vascular endothelial cells” are combined into one network referred to as “Migration of endothelial cells”. All IPA analyses were performed on the 12th March 2020.

### 2.11. Real-Time PCR

TaqMan qPCR primers/probes were used to study mRNA expression of 4 selected angiogenesis-related genes and *GAPDH*. The total reaction volume was 10 μL, consisting of LuminoCt qPCR ready mix (Sigma-Aldrich, St. Louis, USA), TaqMan Primer/Probe (Applied Biosystems, Foster City, USA), water, and cDNA. The cycling condition used was as follows: at 95 °C for 1 s and at 60 °C for 20 s for 40 cycles in addition to one-step initialization at 95 °C for 20 s in QuantStudio 7 Flex Realtime PCR system (Applied Biosystems, Foster City, USA). Then, relative quantities were recorded for each well and normalized to the expression of the housekeeping gene, *GAPDH*.

### 2.12. Olink Proteomics Analyses

Cell culture supernatants and cell lysates from 4 independent experiments were analyzed by Olink Proteomics (Uppsala, Sweden) using Cardiovascular III (CVDIII) and Inflammation (IFN) panels. This platform applies proximity extension assay (PEA) to measure 92 different proteins per panel. Protein amount is reported as normalized protein expression (NPX) on a log2 scale. The NPX values of target protein were compared between untreated samples versus IL-6/sIL-6R-treated (48 h) samples.

### 2.13. Western Blotting

Separation of proteins was achieved by running electrophoresis on cell lysates (10 µg) mixed with SDS-sample buffer after sample denaturation at 95 °C for 5min. NuPAGE Novex Bis-Tris gels (4–12%) and MOPS running buffer were used (both Invitrogen, Carlsbad, USA). MagicMark XP Western Protein Standard (Invitrogen, Carlsbad, USA) was also run together with the samples to determine molecular masses of the protein bands. Proteins were blotted onto Immobilon-FL PVDF membranes (Millipore, Burlington, USA). For further steps, TBS-T (10 mM Tris-HCl (pH 8.0), 150 mM NaCl, 0.1% (w/v) Tween-20) was used. For detecting signaling proteins, we incubated membranes with the following primary antibodies: anti-phospho-AKT^S473^ antibody (Cell Signaling Technology, USA, #4060; 1:2000 dilution), anti-phospho-ERK1/2^T202/Y204^ antibody (Cell Signaling Technology, Danvers, USA, #9106; 1:2000 dilution), anti-phospho-p38^T180/Y182^ antibody (Cell Signaling Technology, USA, #2478; 1:1000 dilution), anti-AKT antibody (Cell Signaling Technology, Danvers, USA, #2920; 1:2000 dilution), anti-ERK1/2 antibody (Cell Signaling Technology, Danvers, USA,#4695; 1:1000 dilution), anti-p38α antibody (Santa Cruz biotechnology, Dallas, USA, #SC-535; 1:750 dilution), and anti-β-tubulin antibody (Millipore, Burlington, USA, #05–661; 1:2000 dilution). This was followed by incubation with horseradish peroxidase (HRP)-conjugated goat anti-rabbit IgGs (Cell Signaling Technology, Danvers, USA, #7074; 1:2000 dilution), horse anti-mouse IgGs (Cell Signaling Technology, Danvers, USA, #7076; 1:2000), or IR-conjugated donkey anti-goat IgGs (Li-Cor Biosciences, Lincoln, USA, #925-68074; 1:10,000 dilution). By briefly covering the PVDF membranes with Immobilon Western Chemiluniescent HRP Substrate solution (Millipore, Burlington, USA), the protein bands were visualized and recorded by a Li-Cor Odyssey Fc imager (Li-Cor Biotechnology UK Ltd., Cambridge, United Kingdom). Further analysis and quantification of the bands was performed using Image Studio Software (Li-Cor Biotechnology UK Ltd., Cambridge, United Kingdom). Stripping of membranes for re-probing was done according to the manufacturer’s instructions using RestorePlus Western blot stripping buffer (Thermo Fisher Scientific, Waltham, USA, #46430).

### 2.14. Data Analysis

Data analysis was performed using GraphPad Prism statistical software version 8.0 (GraphPad Software, Inc., San Diego, USA). Data from at least 3 independent experiments are presented using dot plots and as mean ± standard error of the mean (SEM). For comparison between groups, Student’s *t*-test and ANOVA followed by Bonferroni post-hoc test was used. A *p*-value less than 0.05 was considered as statistically significant.

## 3. Results

### 3.1. Effect of IL-6 Signaling on Tube Formation of Human Vascular ECs

To investigate the effect of IL-6 signaling on the tube formation ability of human ECs, we seeded HUVECs onto a Matrigel and stimulated the cells with IL-6 alone (classic-signaling) or together with sIL-6R (trans-signaling), and we monitored tube formation over a period of 24 h. We found that the tube formation as measured by total tube length, number of loops, and number of branching points was not significantly different in cells stimulated with IL-6 alone or combined with sIL-6R versus unstimulated controls (Appendix A). However, when we pre-treated HUVECs with IL-6 alone or together with sIL-6R for 48h prior to seeding onto a Matrigel, we found that cells treated with the combination of IL-6 and sIL-6R, but not IL-6 alone, showed reduced tube formation, exhibited by decreased total tube length, number of branching points, and loops formed compared to untreated controls (Figure 1A–F). Since ECs release a low level of IL-6 themselves and express the receptors required for IL-6 signaling (i.e., IL-6Rα and gp130), we further investigated the importance of autocrine IL-6 signaling for the ability of ECs to form tubes by silencing the expression of the endogenous IL-6 using stealth siRNAs. As illustrated in Figure 1G–K, ECs with a 90% depletion in endogenous IL-6 production (Appendix A) showed decreased extent of total tube length, number of branching points, and number of loops formed compared to cells exposed to non-target siRNAs. These findings demonstrated that IL-6 trans-signaling impedes the tube formation capacity of vascular ECs while at the same time, the autocrine signaling is essential for proper tube formation ability of the cells.

### 3.2. Regulation of Proliferation and Migration of Human Vascular ECs by IL-6

Next, we investigated the impact of IL-6 on the two crucial processes that are part and parcel of endothelial angiogenic response, i.e., ECs proliferation and migration. To determine whether IL-6 signaling regulates the proliferation of ECs, we stimulated HUVECs with IL-6 alone or in combination with sIL-6R and followed the increase in cell number over 7 days. As presented in Figure 2A, stimulation with a combination of IL-6 and sIL-6R, but not IL-6 alone, significantly reduced the rate of proliferation of human vascular ECs compared to untreated cells. In addition, we assessed whether stimulation with IL-6 and sIL-6R induced apoptosis in ECs using flow cytometry-based 7-AAD/annexin V detection. We found that over 95% of the cells were viable (i.e., negative for both 7-AAD and annexin V) with or without exposure (48 h) to either IL-6 alone or in combination with sIL-6R (Figure 2B). Furthermore, we determined the effect of IL-6 signaling on migration of ECs using a scratch assay on HUVECs stimulated with IL-6 alone or in combination with sIL-6R for 24h. A scratch was made on confluent monolayer of cells and the gap/wound closure was monitored over a period of 12 h. We found that the gap closure by the cells treated with the combination of IL-6 and sIL-6R, but not IL-6 alone, was significantly lower than untreated controls (Figure 2C,D). Overall, these data indicated that IL-6 trans-signaling impairs the proliferation and migration of vascular ECs, but does not affect the viability of the cells.

### 3.3. Exploring Molecular Mechanisms Behind Regulation of Vascular Endothelial Tube Formation and Migration by IL-6 trans-Signaling 

In order to understand the underlying mechanisms for IL-6 trans-signaling-mediated inhibition of endothelial migration, proliferation, and tube formation, we explored alterations in the angiogenesis-related gene expression pattern of human vascular ECs treated with IL-6 in combination with sIL-6R (48 h) compared to unstimulated control using a human angiogenesis gene array. We found that the expression of a total of 17 angiogenesis-associated genes was significantly altered by IL-6 trans-signaling, out of which 10 were downregulated and 7 were upregulated (*p*-value <0.05, FDR = 5%). Appendix A shows the list of genes in the array with respective fold changes, *p*-values, and FDR. To validate the gene expression data from the angiogenesis array, we selected the top two upregulated (CXCL10 and SERPINF1) and downregulated (cKIT and CXCL8) genes that were altered by IL-6 trans-signaling. Using qPCR, we followed the change in expression of these genes in ECs treated with a combination of IL-6 and sIL-6R for 30 min up to 48 h. As demonstrated in Figure 3A,B, the gene expression of both CXCL10 and SERPINF1 significantly increased, particularly in the later time points, while both cKIT and CXCL8 (Figure 3C,D) showed time-dependent decline in their gene expression in response to IL-6 trans-signaling. In addition, the change in gene expression of CXCL10 and CXCL8 by vascular ECs due to activation IL-6 trans-signaling was confirmed at the protein level using PEA (Olink proteomics, Figure 3E,F).

The gene expression data were further analyzed in Ingenuity Pathway Analysis (IPA, QIAGEN Inc., https://www.qiagenbioinformatics.com/products/ingenuity-pathway-analysis) online tool [28], specifically looking at angiogenesis-related networks and functions. As expected, the differentially regulated genes significantly enriched functions related to tube formation of endothelial cells such as “Vasculogenesis” (*p*-value = 1.44E-39), “Formation of endothelial tube” (*p*-value = 4.69E-17), “Tubulation of endothelial cells” (*p*-value = 9.38E-16), and “Neovascularization” (*p*-value = 2.81E-14). Similarly, functions associated with migration of endothelial cells such as “Cell movement of endothelial cells” (*p*-value = 1.42E-41), “Migration of endothelial cells” (*p*-value = 5.04E-34), and “Migration of vascular endothelial cells” (*p*-value = 6.0E-17) were significantly enriched. In line with our functional data (Figure 1A–F), the IPA analyses also predicted that the differential regulation of genes due to IL-6 trans-signaling leads to the inhibition of endothelial tubulation and migration (Figure 4). In summary, our data suggest that IL-6 trans-signaling inhibits endothelial tube formation and migration by upregulating known anti-angiogenic factors such as CXCL10, and by downregulating known pro-angiogenic factors such as CXCL8.

### 3.4. Impact of IL-6 Trans-Signaling on VEGF-A Signaling in Vascular ECs

The IL-6 tans-signaling-induced inhibition of tube formation by ECs was only seen when we used normal Matrigel that contained higher amounts of VEGF-A compared to growth-factor-reduced (GFR) Matrigel (data not shown). This observation led us to hypothesize that activation of IL-6 trans-signaling in ECs might interfere with their response to VEGF-A. As shown in Appendix A, the gene expression of the VEGF-A receptors (*VEGFR1/FLT1* and *VEGFR2/KDR*) on ECs was not altered in response to IL-6 trans-signaling. We further investigated whether IL-6 trans-signaling interfered with VEGF-A signaling by pre-treating HUVECs with IL-6/sIL-6R (48 h), followed by stimulation with VEGF-A for 5 min to 30 min. We found that ECs that were pre-stimulated with IL-6/sIL-6R exhibited reduced Erk1/2^Thr202/Tyr204^, but increased p-p38^T180/Y182^ phosphorylation in response to VEGF-A compared to cells without IL-6/sIL-6R pre-stimulation (Figure 5A,B). The phosphorylation of Akt^Ser473^ in response to VEGF-A was similar in both IL-6/sIL-6R pre-treated and unstimulated cells (Figure 5C). These findings imply that pre-exposure of vascular ECs to IL-6 trans-signaling interferes with VEGF-A signaling and thereby with the angiogenic response of ECs to VEGF-A.

## 4. Discussion

In this study, we investigated the effect of IL-6 signaling in regulating the sprouting angiogenic response of vascular ECs. We demonstrated that activation of IL-6 trans-signaling impaired the migration, proliferation, and tube formation ability of vascular ECs. At the molecular level, we showed that IL-6 trans-signaling in ECs upregulated established anti-angiogenic factors such as *CXCL10* and *SERPINF1* while at the same time downregulating known pro-angiogenic factors such as *cKIT* and *CXCL8*. Furthermore, we revealed that IL-6 trans-signaling exposure of vascular ECs interfered with their response to VEGF-A causing increased p38, but decreased Erk1/2 phosphorylation.

Sprouting angiogenesis is triggered in response to tissue hypoxia due to growth/development and repair/healing, but also due to chronic inflammation [13]. Pro-inflammatory regulators such as TNF-α and IL-1 have been shown to promote angiogenesis [14,15]. The present study was conducted to explore the angiogenic effect of IL-6, a pleiotropic cytokine with pro-inflammatory [16] as well as anti-inflammatory functions [17,18,19]. Previously, an indirect pro-angiogenic effect of IL-6 has been reported from endothelial co-cultures with other cell types such as synovial, peritoneal, and different types of cancer cells [23,24,25,26]. In this study, we provide evidence for the direct angiogenic activity of IL-6 classic- and trans-signaling pathways on ECs using the Matrigel tube formation assay, an in vitro technique widely used to assess the initial phase of sprouting neoangiogenesis including migration and tube formation ability of ECs [29]. We demonstrated that IL-6 trans-signaling impaired the migration and tube formation ability of ECs in addition to inhibiting endothelial proliferation. We also investigated the role of IL-6 classic-signaling because ECs express both the membrane bound IL-6R and gp130 which enable the cells to respond to exogenous as well as endogenous IL-6 [27]. We showed that exposure of vascular ECs to exogenously added IL-6 had a limited effect on their migration, proliferation, or tube formation ability. However, when we depleted the expression of IL-6 in ECs by using siRNAs, we saw impaired tube formation ability of ECs, signifying the crucial role of autocrine IL-6 classic-signaling in their angiogenic response. This is complementary to our recent study that showed the significance of an autocrine IL-6 classic-signaling in basal endothelial functions such as cell proliferation and movement [30]. Taken together, these findings demonstrate that autocrine IL-6 classic-signaling is vital to maintain EC angiogenic function, while activation of IL-6 trans-signaling impairs their proliferation, migration, and tube formation ability.

Angiogenesis is a tightly regulated process by a delicate balance between pro- and anti-angiogenic factors [31,32]. To understand whether IL-6 trans-signaling shifts this balance to inhibit tube formation, we analyzed the gene expression of 92 angiogenesis-related genes in ECs exposed to IL-6 trans-signaling. We found that IL-6 trans-signaling indeed regulated many of the angiogenesis-related genes. As expected, IPA analyses revealed that these differentially regulated genes result in enrichment of functions related to endothelial tube formation and migration. In harmony with our experimental data, the IPA analyses also predicted that the differential regulation of the genes due to IL-6 trans-signaling lead to the inhibition of endothelial tubulation and migration. The top upregulated genes by IL-6 trans-signaling include *CXCL-10* and *SERPINF1*, both of which have been reported to inhibit VEGF-A-induced motility and tube formation of ECs [33,34,35,36]. Simultaneously, IL-6 trans-signaling in ECs downregulates genes including *cKIT* (*CD117*) and *CXCL8*. Signaling through the receptor tyrosine kinase cKIT has been shown to promote survival, migration, and tube formation of ECs [37,38]. CXCL8 is also known to induce endothelial proliferation/migration and tube formation [39,40]. Moreover, it is established that the aforementioned differentially regulated genes mediate their impact on the angiogenic response of ECs in an autocrine manner as the receptors/ligand are expressed by the cells [33,34,35,36,37,38,39,40,41]. Therefore, our data suggest that IL-6 trans-signaling induces endothelial phenotype with reduced tube formation and migration ability by upregulating established endogenous anti-angiogenic factors while at the same time downregulating known endogenous pro-angiogenic factors.

In our hands, the IL-6 trans-signaling inhibited endothelial tube formation in conditions where VEGF-A was present in high concentration. Since VEGF-A is the dominant driver and the rate-limiting factor of angiogenesis [3], we further investigated whether IL-6 trans-signaling interfered with VEGF-A signaling in ECs. We demonstrated that ECs pre-exposed to IL-6 trans-signaling showed increased p38, but decreased Erk1/2 phosphorylation in response to VEGF-A, while Akt phosphorylation remained unaffected. An impaired intracellular signaling of ECs in response to VEGF-A has previously been shown to interfere with their sprouting angiogenic responses [42,43]. Thus, our data imply that IL-6 trans-signaling impairs endothelial tube formation by interfering with VEGF-A signaling.

Collectively, our data demonstrate the dual facets of IL-6 in regulating the sprouting angiogenic function of vascular ECs in which the autocrine IL-6 classic-signaling is vital to maintain this function while activation of IL-6 trans-signaling impairs endothelial proliferation, migration, and tube formation ability. In addition, we shed light on the molecular mechanisms behind IL-6 trans-signaling-mediated impairment of endothelial sprouting angiogenic response, which includes differential regulation of genes involved in angiogenesis and also altering the response of ECs to VEGF-A.

## Figures and Tables

**Figure 1 cells-09-01414-f001:**
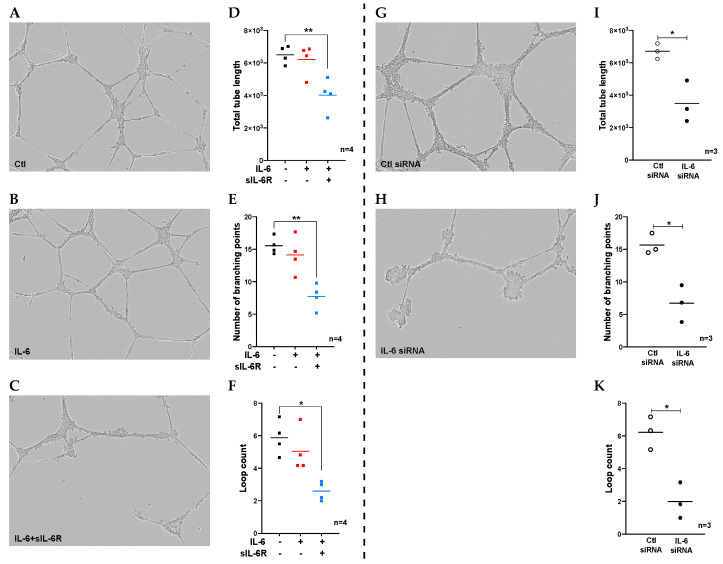
The role of exogeneous and endogenous IL-6 in tube formation of vascular endothelial cells (ECs). Representative images (24h) showing tube formation of (**A**) unstimulated ECs or (**B**) ECs treated with IL-6 alone (100 ng/mL) or (**C**) in combination with sIL-6R (100 ng/mL). Quantification of the tube formation is presented as (**D**) total tube length, (**E**) number of branching points, and (**F**) loop count per image. Tube formation of ECs exposed to non-target siRNA (**G**) or IL-6 targeting siRNAs (**H**) and the quantification of tube formation is shown as (**I**) total tube length, (**J**) number of branching points, and (**K**) loop count per image. Dot plots from 3-4 experiments that were each run in duplicate is presented. * *p* < 0.05, ** *p* < 0.01 compared to control.

**Figure 2 cells-09-01414-f002:**
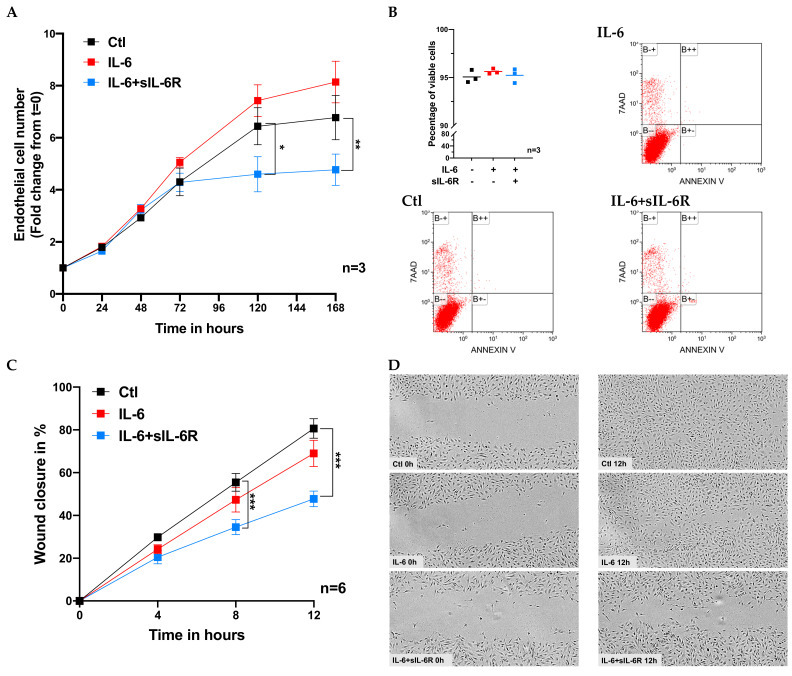
Proliferation and migration of vascular ECs after stimulation with IL-6 (100 ng/mL) and sIL-6R (100 ng/mL). (**A**) The rate of EC proliferation in response to IL-6/sIL6R over a period of 168 h (7 days) is shown as fold change (mean ± SEM) in cell number from baseline time = 0 h (set to 1). (**B**) Scatter plot and flow cytometry data showing the viability of ECs stimulated with IL-6/sIL6R compared to untreated controls and representative density plots are presented. The percentage of wound closure after a scratch assay on vascular ECs pre-treated with IL-6 (100 ng/mL) and sIL-6R (100 ng/mL) is shown as mean ± SEM of 6 experiments each run in duplicate (**C**) and representative images are shown at baseline and after 12 h incubation (**D**). The images are brightness/contrast-adjusted and were cropped using ImageJ. **p* < 0.05, ** *p* < 0.01, *** *p* < 0.001 compared to control. 7-AAD = 7-aminoactinomycin D.

**Figure 3 cells-09-01414-f003:**
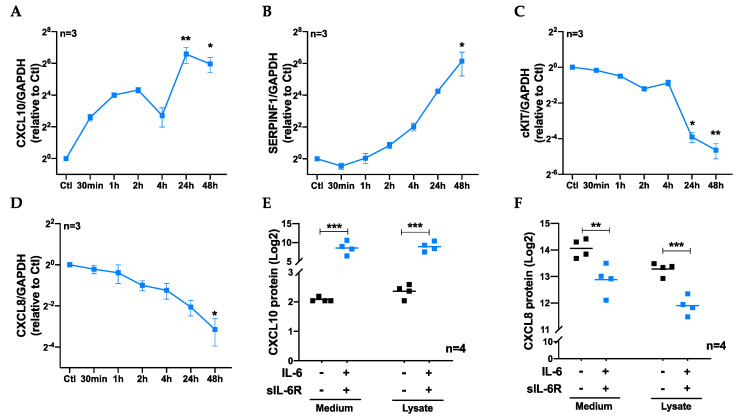
Gene and protein expression of selected genes regulated by IL-6 trans-signaling in vascular ECs. qPCR analyses showing the gene expression (mean ± SEM) of (**A**) CXCL10, (**B**) SERPINF1, (**C**) cKIT, and (**D**) CXCL8 by ECs treated with IL-6 (100 ng/mL) and sIL-6R (100 ng/mL) combined compared to untreated controls (set to 1). The dot plots (mean) show Olink proteomic data on the production of (**E**) CXCL10 and (**F**) CXCL8 by vascular ECs in response to IL-6 (100 ng/mL) and sIL-6R (100 ng/mL) combination (48 h). **p* < 0.05, ** *p* < 0.01, *** *p* < 0.001 compared to control.

**Figure 4 cells-09-01414-f004:**
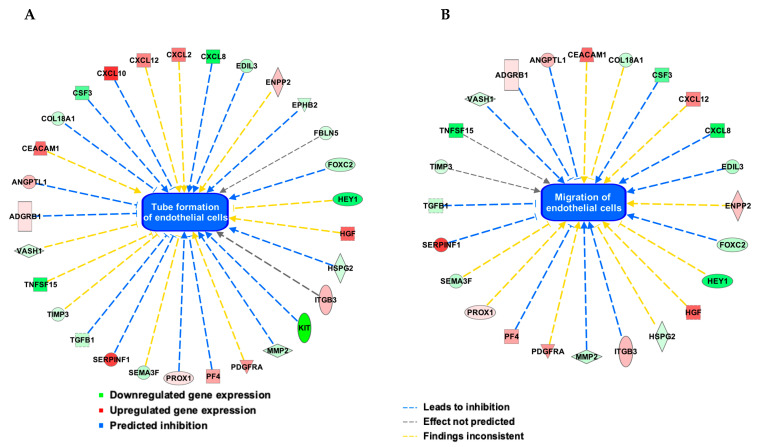
Ingenuity Pathway Analysis (IPA) of the angiogenesis array gene expression data from vascular ECs after treatment with IL-6 (100 ng/mL) and sIL-6R (100 ng/mL) combined (48 h). The differentially regulated genes in response to IL-6 trans-signaling enriched the functions related to tube formation of endothelial cells (**A**) and migration of endothelial cells (**B**). The downregulated and upregulated genes in response to IL-6 trans-signaling are shown with green and red colors, respectively. Inhibition of the functions “tube formation” and “migration” of ECs are depicted in blue color (center), and the differentially regulated genes that lead to the inhibition of tube formation and migration are connected with blue lines. The yellow lines show disagreement between state of the differentially regulated gene expression and the state of tube formation or migration of ECs (i.e., inhibition). Gray lines indicate that no prediction could be made.

**Figure 5 cells-09-01414-f005:**
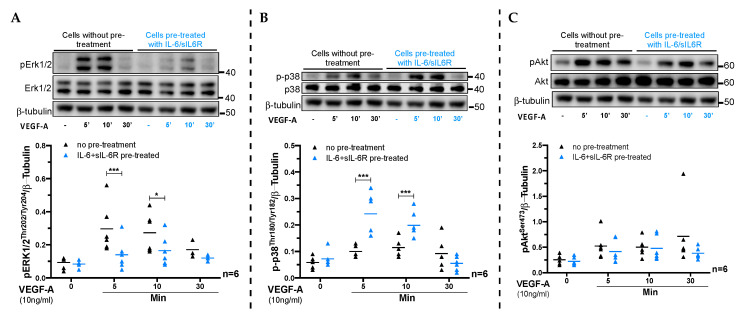
The effect of IL-6 trans-signaling on vascular endothelial growth factor-A (VEGF-A) signaling in vascular ECs. Representative immunoblots (top row) and quantified signal (bottom row) of phosphorylation of (**A**) Erk1/2^Thr202.Tyr204^, (**B**) p38^T180/Y182^, and (**C**) Akt^S473^ in response to VEGF-A (10ng/mL) in ECs with or without IL-6 + sIL-6R (100 ng/mL each) pre-treatment. The dot plots (mean) show signals from the phosphorylated proteins normalized to β-tubulin (loading control) compiled from six independent experiments. * *p* < 0.05, *** *p* < 0.001.

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
