# Peer review of "IL-6 trans-Signaling Impairs Sprouting Angiogenesis by Inhibiting Migration, Proliferation and Tube Formation of Human Endothelial Cells"

_cells, 2020, doi:10.3390/cells9061414_

Round 1
Reviewer 1 Report
In this study the authors postulate that IL-6 trans-signaling impairs sprouting angiogenesis by inhibiting migration, proliferation and tube formation of human endothelial cells. This is an interesting concept that deserves attention but I have some concerns that would like to take into account.
Authors indicate that in their study tube formation was not significantly different in cells stimulated with IL-6 alone or combined with sIL-6R vs unstimulated controls (supplementary figure 1) and experiments got relevancy when pretreated cells 48h before than seed in Matrigel. This set of experiments leads authors to use IL6-siRNA confirming the relevance of endogenous IL6 in the tube formation.
Next, results showed in Figure 2A (proliferation of HUVECs) show that IL-6+sIL-6R present an inhibitory effect 5 days after stimulation, but no effect was observed at 2 days or before. However, wound healing assays are in agreement with the angiogenic and tube formation time line (inhibitory effect observed at 8 hours). In order to understand the in vitro functional processes and underlying mechanisms addressed by Il6 trans signaling, it would be expected a defective proliferative/migrative EC phenotype previous to the impaired angiogenic effect. I would suggest include migration based on TWs assays in order to clarify this point and take into account differences between migration and proliferation within the discussion. In addition, why were not included IL6 silencing experiments in order to elucidate the relevance of endogenous IL6 in migration and/or proliferation?
I missed along the manuscript, data concerning time of mediators incubation to measure levels of protein CXCL10 and 8 (fig E and F). Please detail the time of stimulation in order to link your results with the functional processes involved and your RNA data.
Minnor: please review some spelling details. example Line:326
Reviewer 2 Report
A very interesting, well performed, and clearly written paper with a good methodology. I would recommend acceptance after the minor revision. It significantly extends our knowledge on the pro- and anti-angiogenic role of IL-6 classic- and trans-signaling. In my opinion (as an endothelial biologist), it is of exceptionally high priority.
Figure 2
Figure 2A: please detalise the Y axis labeling (what is the measure of "proliferation of HUVEC"?)
Figure 2B-E: flow cytometry data can be merged into one panel (B).
Figure 2F: change the labeling of X axis to "hours" to be consistent with the labeling principle in Figure 2A.
Figure 2G: equalise all images within the panel by size.
Hereinafter: Please also show P values in a numerical manner instead of using an ascending number of asterisks. Also please replace bar graphs with a box-and-whisker graphs combined with a scatterplot (GraphPad, min to max, show all points).
Figure 3
The Y axis of the fold change is better to be presented at a log scale instead of a linear scale.
Round 2
Reviewer 1 Report
Thank you for your clarifications.